# Association between Serum Phytosterols and Lipid Levels in a Population-Based Study

**DOI:** 10.3390/nu16060775

**Published:** 2024-03-08

**Authors:** Laura Stanasila, Dieter Lütjohann, Julius Popp, Pedro Marques-Vidal

**Affiliations:** 1Department of Medicine, Internal Medicine, Lausanne University Hospital and University of Lausanne, 46 Rue du Bugnon, 1011 Lausanne, Switzerland; laura.stanasila@unil.ch; 2Institute of Clinical Chemistry and Clinical Pharmacology, University Hospital Bonn, 53127 Bonn, Germany; dieter.luetjohann@ukbonn.de; 3Department of Psychiatry, Psychotherapy and Psychosomatics, Psychiatric Hospital, University of Zürich, Lenggstrasse 31, P.O. Box 363, 8032 Zürich, Switzerland; julius.popp@uzh.ch

**Keywords:** phytosterols, blood lipid levels, epidemiology, LDL-cholesterol, HDL-cholesterol

## Abstract

The association between phytosterols and lipid levels remains poorly assessed at a population level. We assessed the associations between serum levels of six phytosterols (campesterol, campestanol, stigmasterol, sitosterol, sitostanol and brassicasterol) and of lipids [total, low-density lipoprotein (LDL)- and high-density lipoprotein (HDL)-cholesterol, triglycerides, apolipopoprotein A-IV and lipoprotein Lp(a)] in two cross-sectional surveys of a population-based, prospective study. Data from 910 participants (59.1% women, 70.4 ± 4.7 years) for the first survey (2009–2012) and from 721 participants (60.2% women, 75.1 ± 4.7 years) for the second survey (2014–2017) were used. After multivariable adjustment, all phytosterols were positively associated with total cholesterol: slope and (95% confidence interval) 1.594 (1.273–1.915); 0.073 (0.058–0.088); 0.060 (0.044–0.076); 2.333 (1.836–2.830); 0.049 (0.033–0.064) and 0.022 (0.017–0.028) for campesterol, campestanol, stigmasterol, sitosterol, sitostanol and brassicasterol, respectively, in the first survey, and 1.257 (0.965–1.548); 0.066 (0.052–0.079); 0.049 (0.034–0.063); 1.834 (1.382–2.285); 0.043 (0.029–0.057) and 0.018 (0.012–0.023) in the second survey, all *p* < 0.05. Similar positive associations were found between all phytosterols and LDL cholesterol. Positive associations were found between campesterol and sitosterol and HDL-cholesterol: slope and (95% CI) 0.269 (0.134–0.405) and 0.393 (0.184–0.602) for campesterol and sitosterol, respectively, in the first survey, and 1.301 (0.999–1.604) and 0.588 (0.327–0.849) in the second survey, all *p* < 0.05. No associations were found between phytosterols and triglyceride or lipoprotein Lp(a) levels, while a positive association between campesterol and apolipoprotein A-IV levels was found: 2.138 (0.454–3.822). Upon normal dietary intakes, serum phytosterol levels were positively associated with total and LDL cholesterol levels, while no consistent association with other lipid markers was found.

## 1. Introduction

Cardiovascular disease (CVD) is the leading cause of death worldwide, accounting for more than one third of total mortality. It is also a major contributor to the increasing burden of health care costs [1]. Dyslipidemia, defined as elevated blood levels of low-density lipoprotein (LDL) and triglycerides and/or lowered levels of high-density lipoprotein (HDL), is one of the modifiable risk factors for CVD. As such, dyslipidemia is the target of many therapeutic strategies, involving drugs as well as dietary supplements. Among the latter, phytosterols have lately received sustained attention [2,3]. Phytosterols are fat-soluble triterpenes found in plant cell membranes, where they play a structural role, similar to cholesterol in animal cells. Their structure only differs from that of cholesterol by a C17-attached side chain. Phytosterols and phytostanols, collectively designated here as PSs, are naturally present in the human diet, the major sources being vegetable oils and cereals, followed by fruit and vegetables [4]. The most common PSs encountered in the Western diet are beta-sitosterol (60%), campesterol and stigmasterol [5].

Despite the lack of evidence of clinical benefits in preventing CVD [6], PSs are accepted as a reliable means to help correct hypercholesterolemia and fight atherosclerosis [7]. An umbrella review of international meta-analyses reported that the administration of 2 g/day of PS decreased total cholesterol, LDL-cholesterol and triglycerides [8], and a large randomized controlled trial showed that the administration of a ready-to-drink dietary supplement with 2.5 g of PS decreased total cholesterol, LDL-cholesterol and apo B100 in a group of patients with polygenic hypercholesterolemia [9]. While most studies have focused on the administration of PS in large doses, little is known about the capacity of dietary PS to influence the blood lipid profile. Daily PS intake averages 250–400 mg [10], similarly to cholesterol, and it can reach 600 mg in vegetarians. Less than 2% of PS is absorbed, vs. 50–60% of cholesterol [11]. This accounts for PS plasma levels being two orders of magnitude lower than cholesterol levels [12]. Interestingly, PS plasma concentrations are not necessarily correlated with intake of usual dietary amounts [10]. A study of PS and cholesterol intake in the Adventist population suggested that the high proportion of PS in their diet contributed to the low levels of plasma cholesterol among non-vegetarians [10]. Other studies found lower levels of LDL in individuals with higher PS intakes in a Swedish population [13], in the Dutch European Prospective Investigation into Cancer and Nutrition (EPIC) [14] and in the Norfolk EPIC cohort [15]. Conversely, in the latter, another study found a positive association between PS concentration and total cholesterol and LDL [16]. Thus, the exact association between PS and lipid levels should be further explored.

In the present study, we aimed to assess the possible association between serum PS levels and concentration levels of LDL, HDL, total cholesterol (TC) and triglycerides in two cross-sectional evaluations of a population-based study in Lausanne, Switzerland.

## 2. Materials and Methods

### 2.1. Population

The CoLaus|PsyCoLaus (https://www.colaus-psycolaus.ch) is a prospective cohort study established in 2003 following every 5 years a sample of the inhabitants of the city of Lausanne (Switzerland), aged 35 to 75 years at baseline [17]. In each survey, participants answered questionnaires, underwent clinical examination and blood samples were drawn for analyses. Recruitment began in June 2003 and ended in May 2006; the first follow-up (FU1) was performed between April 2009 and September 2012 and the second follow-up (FU2) was performed between May 2014 and April 2017. Sterol assessment was performed in the first and second follow-ups.

### 2.2. Sterol Assessment

In both follow-ups, we measured serum concentrations for four plant sterols (sitosterol, campesterol, stigmasterol and brassicasterol) and two plant stanols (sitostanol and campestanol).

Serum phytosterols were assessed in the Institute of Clinical Chemistry and Clinical Pharmacology, University Hospital Bonn, Bonn, Germany, as indicated previously [18,19]. Briefly, gas chromatographic separation and the detection of cholesterol and 5α-cholestane (ISTD) were performed on a DB-XLB 30 m × 0.25 mm i.d. × 0.25 μm film thickness (J&W Scientific Alltech, Folsom, CA, USA) in a Hewlett-Packard (HP) 6890 Series GC-system (Agilent Technologies, Palo Alto, CA, USA), equipped with an FID.

Sterols and oxysterols were separated on another DB-XLB column (30 m × 0.25 mm i.d. × 0.25 μm film thickness, J&W Scientific Alltech, Folsom, CA, USA) in a HP 6890 N Network GC system (Agilent Technologies, Waldbronn, Germany) connected with a direct capillary inlet system to a quadruple mass selective detector HP5975B inert MSD (Agilent Technologies, Waldbronn, Germany). Both GC systems were equipped with HP 7687 series auto samplers and HP 7683 series injectors (Agilent Technologies, Waldbronn, Germany).

To avoid autoxidation, 50 μL of a 2.6.-di-tert.-butylmethylphenol/methanol solution (mg/mL) (Sigma-Aldrich Chemie GmbH, Taufkirchen, Germany) was added to the samples. After saponification with 2 mL 1 M 95% ethanolic sodium hydroxide solution (Merck KGaA, Darmstadt, Germany) at 60 °C for one hour, the free sterols and oxysterols were extracted three times with 3 mL cyclohexane each. The organic solvent was evaporated by a gentle stream of nitrogen at 60 °C on a heating block. The residue was dissolved in 80 μL n-decane (Merck KGaA, Darmstadt, Germany). An aliquot of 40 μL was incubated (1 h at 70 °C on a heating block) with the addition of 20 μL of the trimethylsilylating (TMSi) reagent (chlortrimethylsilane (Merck KGaA, Darmstadt, Germany)/1.1.1.3.3.3-Hexamethyldisilasane (Sigma Aldrich, Co., St. Louis, MO, USA)/pyridine (Merck KGaA, Darmstadt, Germany), 9:3:1) in a GC vial for GC-MSD non-cholesterol and oxysterol analysis. Another aliquot of 40 μL was incubated with the addition of 40 μL of the TMSi-reagent and dilution with 300 μL n-decane in a GC vial for GCFID cholesterol analysis.

An aliquot of 2 μL was injected using automated injection in a splitless mode using helium (1 mL/min) as carrier gas for GC–MS-SIM and hydrogen (1 mL/min) for GC-FID analysis at an injection temperature of 280 °C. The temperature program for GC was as follows: 150 °C for three minutes, followed by 20 °C/min up to 290 °C keeping for 34 min. For MSD electron impact ionization was applied with 70 eV. SIM was performed by cycling the quadruple mass filter between different m/z at a rate of 3.7 cycles/sec.

Peak integration was performed manually. Non-cholesterol sterols were quantified from the ratios of the areas under the curve of the respective non-cholesterol sterols after SIM analyses against internal standards using standard curves for the listed sterols. The identity of all sterols was proven by comparison with the full-scan mass spectra of authentic compounds.

### 2.3. Lipid Markers

Participants were asked to fast at least six hours before blood drawing. Biological assays were performed at the Centre Hospitalier Universitaire Vaudois (CHUV) Clinical Laboratory on fresh blood samples within two hours of blood collection. The methods and corresponding maximum inter- and intra-batch coefficients of variation were the following: total cholesterol (TC) by cholesterol oxidase—phenol 4-aminoantipyrine peroxidase (CHOD-PAP) (1.6–1.7%); HDL-cholesterol by CHOD-PAP + Polyethylene glycol (PEG) + cyclodextrin (3.6–0.9%) and triglycerides by glycerol phosphate oxidase-phenol 4-aminoantipyrine peroxidase GPO-PAP (2.9–1.5%). LDL-cholesterol was assessed using the Friedewald formula. Contrary to total cholesterol assessed using GC, total cholesterol assessed using traditional methods does not differentiate between true cholesterol and other sterol molecules.

In the first follow-up, apolipoprotein A-IV (apoA-IV) and Lp(a) were also assessed. Apolipoprotein A-IV was assessed with a double-antibody enzyme-linked immunosorbent assay using an affinity-purified polyclonal rabbit anti-human apo lipoprotein A-IV antibody for coating and the same antibody coupled to horseradish peroxidase for detection. Plasma containing a known concentration of apolipoprotein A-IV served as the calibration standard. Both the intra- and inter-assay coefficients of variation were below 5%. Lipoprotein (a) was assessed using a double-antibody enzyme-linked immunosorbent assay (ELISA) using an affinity-purified polyclonal apo[a] antibody for coating and the horseradish peroxidase (HRP)-conjugated monoclonal 1A2 for detection and immunoblot. Both apolipoprotein A-IV and Lp(a) assessments were performed in the laboratory of the Division of Genetic Epidemiology, Innsbruck, Austria.

### 2.4. Other Covariates

Smoking status was self-reported and categorized as never, former or current. Participants reported prescribed and over the counter drugs that they were currently taking; statins and ezetimibe were considered. Diabetes was defined as fasting plasma glucose ≥ 7.0 mmol/L or taking antidiabetic drugs. Body weight and height were measured with participants barefoot and in light indoor clothes. Body weight was measured in kilograms to the nearest 100 g using a Seca^®^ scale (Hamburg, Germany). Height was measured to the nearest 5 mm using a Seca^®^ (Hamburg, Germany) height gauge. Body mass index (BMI) was computed and categorized as normal (BMI < 25 kg/m^2^), overweight (25–29.9 kg/m^2^) or obese (≥30 kg/m^2^). No specific dietary survey was conducted to obtain dietary phytosterol intake.

### 2.5. Statistical Analysis

Statistical analyses were conducted using Stata v.16.1 (Stata Corp, College Station, TX, USA) separately for each survey. Gaussian distribution was assessed by visually assessing the distribution of the variable in histograms and linearity of the QQ plot, using the sixplot command of Stata. Results were expressed as number of participants (percentage) for categorical variables and as average (±standard deviation) or median [interquartile range] for continuous variables. Bivariate associations were computed using the Spearman nonparametric test. Multivariable analyses were conducted using linear regression and results were expressed as multivariable-adjusted slope and 95% confidence interval (CI). For multivariable analyses, adjustments were performed on age (continuous), sex (male, female), BMI (continuous), diabetes (yes, no) and statin use (yes, no). Triglycerides and Lp(a) were log-transformed before analysis. A second set of analyses was conducted after excluding participants taking lipid-lowering drugs, and a third set of analyses was conducted stratifying on sex. Statistical significance was considered for a two-sided test with *p* < 0.05.

## 3. Results

### 3.1. Sample Characteristics and Phytosterol Concentrations

The samples assessed in the two follow-ups are described in Table 1. They consist of 910 and 721 individuals, respectively, with an average age of 70.4 (FU1) and 75.1 (FU2) years. Most participants were women, one out of five was obese, less than 15% were current smokers, and one fifth (FU1) and one quarter (FU2) of them took lipid-lowering drugs.

The concentrations of PS are reported in Table 2. Phytosterol levels were similar from one follow-up to the other. Campesterol and sitosterol were the most abundant, the other phytosterol levels being one to two orders of magnitude lower. Women tended to present higher levels than men, but the differences were not consistent between surveys (Appendix A).

### 3.2. Association between Serum Lipids and Phytosterols—Bivariate Analysis

Table 3 displays the bivariate correlations between serum phytosterols and lipid levels. Positive correlations were found between all phytosterols and TC, total cholesterol-GC, and LDL in both follow-ups. Positive correlations were found between most plant sterols and stanols with HDL-cholesterol levels, and this association was consistent between follow-ups. Negative correlations were found between most plant sterols and stanols with triglyceride levels, but those associations were not consistent between follow-ups. Similar findings were obtained when the analysis was restricted to participants devoid of lipid-lowering drug treatment (Appendix A) or when stratifying by sex, although some correlations were no longer significant due to decreased sample size (Appendix A).

### 3.3. Association between Serum Lipids and Phytosterols—Multivariable Analysis

Table 4 shows the results of the multivariable regression analysis between serum phytosterol levels and TC, total cholesterol-GC, HDL, LDL, and triglycerides (log-transformed) as dependent variables in the two follow-ups. All phytosterols were positively associated with TC, total cholesterol-GC and LDL, and this association was consistent between follow-ups. Regarding HDL-cholesterol levels, positive associations were found for campesterol and sitosterol, while for the other phytosterols, associations were small and inconsistent. Regarding triglyceride levels, inconsistent associations were found. Similar findings were obtained when the analysis was restricted to participants devoid of lipid-lowering drug treatment (Appendix A) or after stratifying by sex (Appendix A).

### 3.4. Association with Apolipoprotein A-IV and Lp(a)

The bivariate correlations between phytosterols and apolipoprotein A-IV and Lp(a) are summarized in Table 3. Campesterol, stigmasterol and sitosterol levels were positively correlated with apolipoprotein A-IV, while no correlation was found with Lp(a).

Table 5 shows the results of the association between phytosterols and apolipoprotein A-IV and Lp(a) in FU1. Campesterol levels were positively associated with apolipoprotein A-IV, while no association was found with Lp(a). When the analysis was restricted to participants devoid of lipid-lowering drug treatment, no associations were found with apolipoprotein A-IV or Lp(a) (Appendix A), while when the analysis was stratified by sex, significant associations were found between campesterol and sitosterol with Apo-IV in males (Appendix A).

## 4. Discussion

In the present study, we used data collected from the CoLaus cohort to assess the relationship between serum phytosterol levels and lipid values in a community-dwelling Western European population, in two cross-sectional evaluations at a 5-year interval. Our results indicate that at normal dietary intakes, phytosterol levels are positively associated with total and LDL cholesterol levels, while no consistent associations with other lipid markers were found.

### 4.1. Association with Serum Lipids

In the present study, our results do not support an inverse association between phytosterol and cholesterol levels, whether TC, LDL or HDL. On the contrary, the multivariable analysis adjusting for potential confounders showed a positive association between phytosterols and TC and LDL levels. Our findings agree with a previous study [16], which found the same positive association between serum phytosterol levels and TC and LDL. Conversely, other studies reported an inverse association between phytosterol and TC and LDL levels [13,14,15], but had measured phytosterols intake, not serum levels.

Our multivariable regression showed a lack of association between phytosterols and HDL levels. This agrees with previously published data indicating that doses of 2–3 g/day of phytosterols failed to affect HDL levels [20].

We also found no association between phytosterol and triglyceride levels after the multivariable analysis, although a significant inverse correlation between campesterol, sitostanol and triglyceride levels was found. A previous meta-analysis showed an inverse association between phytosterol intake and triglycerides, depending on baseline triglyceride levels [20], but the included studies used phytosterol doses ranging from 0.8 to 4 g/day, largely superior to usual dietary phytosterol intake and therefore also to the phytosterol levels in the present study.

### 4.2. Association with Apolipoprotein A-IV and Lp(a)

Campesterol, campestanol and sitosterol were positively correlated with apolipoprotein A-IV levels, but this association was no longer significant after multivariable adjustment. Apolipoprotein A-IV is secreted by the small intestine, present in chylomicron remnants and in HDL, and is linked to dietary fat absorption [21]. A study conducted in 2002 concluded that the absorption of phytosterols was independent of apolipoprotein A-IV polymorphism, but a previous meta-analysis of the effects of phytosterol supplementation on apolipoprotein levels provided no information regarding apolipoprotein A-IV [22]. Hence, whether phytosterol intake (and levels) affect this apolipoprotein remain to be assessed.

No association was found between serum phytosterol levels and Lp(a). This was expected, as the expression of Lp(a) is genetically determined and its levels are stable throughout the lifetime [23]. This lack of association would have been found had Lp(a) been measured in both follow-ups. One study involving controlled feeding of a plant-based diet for four weeks showed a downregulation of Lp(a) expression [24]. However, under the dietary range conditions of the present study, such an effect was not demonstrated.

### 4.3. Phytosterols, Plant-Based Diets and Cardiovascular Risk

Plant-based diets, such as the Dietary Approaches to Stop Hypertension (DASH), Mediterranean or vegetarian diets, are well known for their ability to protect against hypertension and support cardiovascular health [25,26]. Diets rich in foods of plant origin were shown to be linked to a better lipid profile and a lower risk of atherosclerosis [27]. In view of the well-established cardioprotective effect of plant-based diets, our results, that show a consistent, albeit weak, positive association between the serum levels of phytosterols and TC, total cholesterol-GC and LDL, might seem surprising and hard to account for. It is worth noting that phytosterol serum levels are not accurately reflective of plant food intake [10]. Also, the major sources of phytosterols in the Western diet are primarily vegetable oils and cereals [5,28], much less than fruits and vegetables that are proven to be heart-healthy and linked to reduced mortality [29]. Indeed, the diet of CoLaus|PsyCoLaus participants came largely short of the recommendations of the Swiss Society of Nutrition regarding fruit and vegetable intake [30]. Unfortunately, it was not possible to evaluate the main sources of PS in the diet of the CoLaus|PsyCoLaus participants, as no PS compositional table was available. The results of the present study tend to indicate that any protective effects of plant-based diets against CVD and more particularly against atherosclerosis are likely to be due to food components other than phytosterols.

### 4.4. Strengths and Limitations

One of the strengths of this study is that it measured a wide array of phytosterols in a large sample of a community-dwelling population. Furthermore, two different follow-ups were performed, which enabled to replicate the findings.

There are several limitations to this study. First, normally occurring phytosterol serum levels in this study fell within a narrow range, as compared to variations achieved in interventional studies when using phytosterol-enriched foods. Still, the values were comparable to those obtained in other countries [10,16], thus suggesting that the ranges observed might well fall within physiological values. Second, the study was conducted in a geographically limited region, and it has been shown that dietary intakes vary between regions in Switzerland [31]. Hence, results might not be replicable in other settings, and it would be interesting to conduct similar studies in other countries or locations. Third, triglycerides were measured without blanking for free glycerol, which could overestimate the values. Still, the correlation coefficient between blanked and non-blanked for free glycerol was 0.99 [32], and as the correlation coefficient is insensitive to arithmetic changes in one variable such as addition or multiplication, the associations observed should not change significantly. Lastly, current clinical laboratory cholesterol measurement techniques do not differentiate between cholesterol and other sterols; hence, the associations observed with total cholesterol as determined by CHOD-PAP might be overestimated as phytosterols are also considered as “cholesterol” in clinical practice. Still, the amount of phytosterols was very low compared to total cholesterol, and associations observed with total cholesterol as determined by gas chromatography–mass spectrometry–selected ion monitoring were similar or even stronger. Hence, the use of current clinical methods to assess the associations between lipids and PS might be adequate.

## 5. Conclusions

Upon normal dietary intake, serum phytosterol levels are positively associated with total and LDL cholesterol levels. Campesterol and sitosterol are associated with HDL cholesterol, while no association was found between circulating phytosterols and triglycerides, lipoprotein Lp(a) and apolipoprotein A-IV.

## Figures and Tables

**Table 1 nutrients-16-00775-t001:** Sample characteristics, CoLaus|PsyCoLaus study, Lausanne, Switzerland.

	First Follow-Up	Second Follow-Up
Sample size	910	721
Women (%)	538 (59.1)	434 (60.2)
Age (years)	70.4 ± 4.7	75.1 ± 4.7
BMI (kg/m^2^)	26.7 ± 4.7	26.5 ± 4.6
BMI categories (%)		
Normal	348 (38.2)	290 (40.2)
Overweight	373 (41.0)	295 (40.9)
Obese	189 (20.8)	136 (18.9)
Smoking status (%)		
Never	370 (40.7)	305 (42.3)
Former	406 (44.6)	330 (45.8)
Current	134 (14.7)	86 (11.9)
Diabetes (%)	154 (16.9)	100 (13.9)
Lipid-lowering drugs (%)		
Statins	189 (20.8)	179 (24.8)
Ezetimibe	7 (0.8)	7 (1.0)
Lipid values		
Total cholesterol (mmol/L)	5.8 ± 1.1	5.3 ± 1.0
HDL cholesterol (mmol/L)	1.7 ± 0.5	1.7 ± 0.5
LDL cholesterol (mmol/L)	3.5 ± 1.0	3.0 ± 0.9
Triglycerides (mmol/L)	0.4 [1.6–0.48]	0.4 [1.4–0.48]
Apolipoprotein A-IV (mg/dL)	17.6 ± 5.0	NA
Lp(a) (mg/dL)	13.4 [6.4–37.0]	NA

BMI, body mass index; HDL, high-density lipoprotein; LDL, low-density lipoprotein; Lp(a), lipoprotein(a); NA, not assessed. Data for the first (2009–2012) and second (2014–2017) follow-ups. Results are expressed as number of participants (column percentage) for categorical variables and as average ± standard deviation or as median [interquartile range] for continuous variables.

**Table 2 nutrients-16-00775-t002:** Serum sterol concentrations, CoLaus|PsyCoLaus study, Lausanne, Switzerland.

	First Follow-Up, n = 910		Second Follow-Up, n = 721	
	Average ± SD	Median [IQR]	Average ± SD	Median [IQR]
Campesterol [mg/dL]	0.32 ± 0.20	0.28 [0.19–0.41]	0.29 ± 0.15	0.26 [0.18–0.37]
Campestanol [µg/dL]	6.05 ± 4.15	5.16 [3.28–7.52]	3.94 ± 1.39	3.71 [3.07–4.54]
Stigmasterol [µg/dL]	6.31 ± 4.00	5.28 [3.34–8.26]	7.85 ± 3.57	7.10 [5.46–9.25]
Sitosterol [mg/dL]	0.25 ± 0.13	0.22 [0.16–0.31]	0.25 ± 0.12	0.22 [0.17–0.31]
Sitostanol [µg/dL]	7.25 ± 4.15	6.23 [4.56–8.83]	4.09 ± 1.86	3.76 [3.27–4.46]
Brassicasterol [µg/dL]	19.4 ± 11.0	17.2 [11.6–24.1]	20.7 ± 10.7	18.5 [13.5–24.9]
Total cholesterol GC [mg/dL]	210 ± 40	209 [184–237]	188 ± 38	191 [163–214]

SD, standard deviation; IQR, interquartile range. Data for the first (2009–2012) and second (2014–2017) follow-ups. Results are expressed as average ± standard deviation and median [interquartile range].

**Table 3 nutrients-16-00775-t003:** Correlation coefficients between serum phytosterols and lipid markers, CoLaus|PsyCoLaus study, Lausanne, Switzerland.

	Total Cholesterol		Total Cholesterol GC		LDL Cholesterol		HDL Cholesterol		Triglycerides		ApoA-IV	Lp(a)
	First	Second	First	Second	First	Second	First	Second	First	Second	First	First
Campesterol	**0.323**	**0.253**	**0.341**	**0.222**	**0.253**	**0.155**	**0.266**	**0.297**	**−0.066**	**−0.128**	**0.087**	0.039
Campestanol	**0.344**	**0.265**	**0.374**	**0.238**	**0.300**	**0.213**	**0.138**	**0.159**	**0.071**	−0.010	0.031	0.057
Stigmasterol	**0.225**	**0.247**	**0.238**	**0.204**	**0.173**	**0.182**	**0.118**	**0.212**	0.043	**−0.103**	**0.083**	0.031
Sitosterol	**0.300**	**0.261**	**0.321**	**0.241**	**0.226**	**0.172**	**0.268**	**0.302**	**−0.077**	**−0.153**	**0.092**	0.041
Sitostanol	**0.217**	**0.176**	**0.249**	**0.151**	**0.170**	**0.115**	**0.106**	**0.150**	**0.066**	−0.016	0.065	0.021
Brassicasterol	**0.242**	**0.233**	**0.257**	**0.217**	**0.194**	**0.157**	**0.145**	**0.207**	0.024	−0.048	0.071	0.019

Results are expressed as Spearman nonparametric correlation coefficient values. Data for the first (2009–2012) and second (2014–2017) follow-ups. Significant (*p* < 0.05) results are indicated in bold.

**Table 4 nutrients-16-00775-t004:** Multivariable regression analysis between lipid and lipoprotein levels (dependent variable) and serum sterol levels, CoLaus|PsyCoLaus study, Lausanne, Switzerland.

	Total Cholesterol		Total Cholesterol-GC		LDL Cholesterol		HDL Cholesterol		Triglycerides	
	First	Second	First	Second	First	Second	First	Second	First	Second
Campesterol	**1.594** **(1.273; 1.915)**	**1.257** **(0.965; 1.548)**	**59.8** **(47.8; 71.9)**	**64.0** **(47.8; 80.2)**	**1.343** **(0.955; 1.730)**	**1.915** **(1.499; 2.330)**	**0.269** **(0.134; 0.405)**	**1.301** **(0.999; 1.604)**	0.062 (−0.086; 0.209)	−0.150 (−0.353; 0.054)
Campestanol	**0.073** **(0.058; 0.088)**	**0.066** **(0.052; 0.079)**	**3.092** **(2.536; 3.648)**	**6.617** **(4.856; 8.379)**	**0.169** **(0.127; 0.210)**	**0.207** **(0.162; 0.252)**	−0.003 (−0.009; 0.004)	0.023 (0.002; 0.045)	**0.016** **(0.009; 0.023)**	0.020 (−0.002; 0.042)
Stigmasterol	**0.060** **(0.044; 0.076)**	**0.049** **(0.034; 0.063)**	**2.246** **(1.646; 2.845)**	**1.947** **(1.243; 2.65)**	**0.052** **(0.036; 0.069)**	**0.063** **(0.045; 0.081)**	0.003 (−0.003; 0.010)	0.009 (0.001; 0.018)	**0.012** **(0.005; 0.019)**	0.001 (−0.008; 0.010)
Sitosterol	**2.333** **(1.836; 2.830)**	**1.834** **(1.382; 2.285)**	**89.1** **(70.5; 107.7)**	**79.0** **(57.7; 100.3)**	**1.757** **(1.251; 2.263)**	**2.264** **(1.713; 2.815)**	**0.393** **(0.184; 0.602)**	**0.588** **(0.327; 0.849)**	0.117 (−0.111; 0.345)	−0.238 (−0.505; 0.029)
Sitostanol	**0.049** **(0.033; 0.064)**	**0.043** **(0.029; 0.057)**	**2.207** **(1.632; 2.783)**	**1.944** **(0.600; 3.289)**	**0.055** **(0.024; 0.087)**	**0.070** **(0.036; 0.105)**	−0.003 (−0.010; 0.003)	0.011 (−0.006; 0.027)	**0.014** **(0.007; 0.021)**	0.006 (−0.010; 0.022)
Brassicasterol	**0.022** **(0.017; 0.028)**	**0.018** **(0.012; 0.023)**	**0.840** **(0.624; 1.055)**	**0.932** **(0.705; 1.159)**	**0.017** **(0.012; 0.023)**	**0.025** **(0.019; 0.031)**	0.002 (0.000; 0.005)	**0.007** **(0.004; 0.010)**	**0.003** **(0.001; 0.006)**	0.000 (−0.003; 0.003)

Results are expressed as slope and (95% confidence interval). Data for the first (2009–2012) and second (2014–2017) follow-ups. Statistical analysis conducted using linear regression adjusting for age (continuous), sex (male, female), BMI (continuous), diabetes (yes, no) and statin use (yes, no). Significant (*p* < 0.05) associations are indicated in bold.

**Table 5 nutrients-16-00775-t005:** Multivariable regression analysis between apolipoprotein A-IV, Lp(a) (dependent variable) and serum sterol levels, CoLaus|PsyCoLaus study, Lausanne, Switzerland.

	Apolipoprotein A-IV	Lp(a), Log-Transformed
Campesterol	**2.138 (0.454; 3.822)**	0.036 (−0.370; 0.442)
Campestanol	0.015 (−0.064; 0.093)	0.007 (−0.012; 0.026)
Stigmasterol	0.027 (−0.056; 0.110)	−0.001 (−0.021; 0.019)
Sitosterol	2.299 (−0.301; 4.899)	0.044 (−0.581; 0.670)
Sitostanol	0.051 (−0.028; 0.130)	0.001 (−0.018; 0.020)
Brassicasterol	0.026 (−0.004; 0.055)	0 (−0.007; 0.007)

Results are expressed as slope and (95% confidence interval). Data for the first follow-up (2009–2012). Statistical analysis conducted using linear regression adjusting for age (continuous), sex (male, female), BMI (continuous), diabetes (yes, no) and statin use (yes, no). Significant (*p* < 0.05) associations are indicated in bold.

## Data Availability

The data of CoLaus|PsyCoLaus study used in this article cannot be fully shared as they contain potentially sensitive personal information on participants. According to the Ethics Committee for Research of the Canton of Vaud, sharing these data would be a violation of the Swiss legislation with respect to privacy protection. However, coded individual-level data that do not allow researchers to identify participants are available upon request to researchers who meet the criteria for data sharing of the CoLaus|PsyCoLaus Datacenter (CHUV, Lausanne, Switzerland). Any researcher affiliated to a public or private research institution who complies with the CoLaus|PsyCoLaus standards can submit a research application to research.colaus@chuv.ch or research.psycolaus@chuv.ch. Proposals requiring baseline data only will be evaluated by the baseline (local) Scientific Committee (SC) of the CoLaus and PsyCoLaus studies. Proposals requiring follow-up data will be evaluated by the follow-up (multicentric) SC of the CoLaus|PsyCoLaus cohort study. Detailed instructions for gaining access to the CoLaus|PsyCoLaus data used in this study are available at https://www.colaus-psycolaus.ch/professionals/how-to-collaborate/, accessed on 5 March 2024.

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
