# Peer review of "Association between Serum Phytosterols and Lipid Levels in a Population-Based Study"

_nutrients, 2024, doi:10.3390/nu16060775_

Round 1
Reviewer 1 Report (Previous Reviewer 2)
Comments and Suggestions for Authors
Dear Editor,
I carefully read the revised version of the manuscript, that is significantly improved compared to the original version.
Author Response
Thank you very much
Reviewer 2 Report (Previous Reviewer 3)
Comments and Suggestions for Authors
Authors well responded and revised the manuscript properly.
Comments on the Quality of English LanguageThere are still some minor grammatical/style issues that a copy editor can deal with.
Author Response
Thank you very much. We tried to correct as much as possible all the typos
Reviewer 3 Report (New Reviewer)
Comments and Suggestions for Authors
Review for the manuscript
Association Between Serum Phytosterols and Lipid Levels in a Population-Based Study
Thank you for inviting me to review this manuscript. I have some suggestions.
OVERALL COMMENTS
This manuscript intentend to “to assess the possible associ- 72 ation between serum PS levels and concentration levels of LDL, HDL, total cholesterol 73 (TC) and triglycerides in two cross-sectional evaluations of a population-based study in 74 Lausanne, Switzerland”. This is an interesting research.
TITLE
It is adquate
ABSTRACT
I suggest reducing the Abstract. Please see tha in line 30 we see: “…survey, all p<0.05. . No associations were found between phytostero><0.05.. No associations….”. Please correct the mistake.
KEYWORDS
I suggest include more keywords such as LDL-c, HDL-c…
INTRODUCTION
This section is adequate, however, I suggest including more references regarding this topic. Please, visit PUBMED.com and include new studies published in 2023-2024.
METHODS, RESULTS and DISCUSSION
These sections are well-performed and have good quality. However, I miss the inclusion of newer references in the Discussion..
CONCLUSION
This section is too short. Please improve. You have good results.
REFERENCES
As suggested before, please, include more references in the Introduction and Discussion sections.
Comments on the Quality of English LanguageMinor corrections are necessary.
Author Response
Keywords: the number of keywords is limited to five. We added LDL-cholesterol; and HDL-cholesterol, but we cannot add more.
Abstract: we checked the statements and they are correct.
Introduction and discussion: The editor requested to remove a large number of references. Hence, we cannot include more references.
Conclusion: we changed the conclusion to. "Upon normal dietary intake, serum phytosterols levels are positively associated with total and LDL cholesterol levels. Campesterol and sitosterol are associated with HDL cholesterol, while no association was found between circulating phytosterols and triglycerides, lipoprotein Lp(a) and apolipoprotein A-IV"
This manuscript is a resubmission of an earlier submission. The following is a list of the peer review reports and author responses from that submission.
Round 1
Reviewer 1 Report
Comments and Suggestions for Authors
The authors try to assess associations between serum phytosterols, i.e. plant sterols (campesterol, brassicastrol, sitosterol, stigmasterol) and plant stanols (5a-campestanol, 5a-sitostanol) levels, and serum lipid levels such as total, LDL-, and HDL-cholesterol as well as apoA-IV and Lp(a). Their main finding is that there is a positive association between serum phytosterol levels (sterols and 5a-stanols) and total and LDL-cholesterol.
Major points:
The authors avoid to explain some important basics with respect to the compared serum parameters. Total cholesterol means the sum of LDL-C plus HDL-C plus VLDL-C plus cholesterol in chylomicrons and chylomicron remnants. Lipoproteins are the transport media for free and esterified cholesterol, triglycerides and xenosterols (for example phytosterols – sterols and 5a-stanols), cholesterol precursors and oxidative metabolites of all sterols. ApoA-IV as an apolipoprotein is associated with HDL and chylomicrons - its function is not so clear. Lp(a) is a lipoprotein resembling LDL-Cholesterol, however presenting other function.
I) It is not clear, why the authors compare these parameters, especially triglycerides, ApoA-IV and Lp(a) with phytosterols. Why not Apo-AI or Apo-B100? In addition it is clear that, as long as campesterol correlates with sitosterol, campesterol with its D22-unsaturated analogue brassicasterol, sitosterol with its D-22-unsatureated analogue stigmasterol, and the completely hydrogenated products of both campesterol and brassicasterol – 5a-campestanol as well as the completely hydrogenated product of both sitosterol and stigmasterol - 5a-sitostanol, all plant sterols and stanols have to correlate with total cholesterol and LDL-cholesterol. This under the assumption that the subjects do not have severe genetic disturbances in lipoprotein and phytosterol metabolism such as homozygous familial sitosterolemia.
The authors conclude: „Upon normal dietary intakes, serum phytotosterol levels were positively associated with total and LDL cholesterol levels.“
This final finding is not at all surprising. Changes (increase or decrease) in LDL particle number and consecutively in total cholesterol are followed by changes (increase and decrease) of total cholesterol, non-cholesterol sterols, 5a-stanols and oxysterols. An exception are the triglycerides, because there are triglycerid-rich and -poor lipoproteins and therefore the TG-content in serum cannot correlate with the cholesterol and non-cholesterol sterol (phytosterols and cholesterol precursors and oxysterols) concentrations.
Because of these coherences, since about 25 years, absolute serum or plasma amounts of total plant sterols and stanols are corrected for the total serum or plasma cholesterol content in or from the same sample. This was already understood and performed by Stanasila and Marques-Vidal in a previous publication in Nutrition 2022, 14,2500, where they used the same data set and even identical mean and median data as in the actual present manuscript from a first (2009-2012) and second (2014-2017) follow-up of the „CoLaus/PsycoLaus study“, Lausanne, Switzerland. Unfortunately, in the previously published manuscript the data for total cholesterol (correction factor or denominator) are not presented. However, calculating the average cholesterol in the first trial indicated in the previous publication, the content from the given ratio data and absolute phytosterol data leads to an average concentration of 227.6 mg/dL, i.e. 5.8 mmol/L, the same amount as given in Table 1 of the actual manuscript under revision now.
This and the fact that some of the average +/- SD levels as well as the Median [IQR] numbers are in part absolute identical and in part nearly identical in the present mansucript (Table 2) compared with the given data in the former publication (Nutrients 2022, Table 2) is astonishing, above all that the number of participants are differing.
Nutrients 2022: First (n=730) Second (n=526)
Actual manuscript: First (n=910) Secon (n=721)
Unfortunately and interestingly, the authors indicate no accessibility to the raw data because of the sensitivity of the data.
Minor points
1) 2.2 Sterol assessment
The internet gives information that the Department of Clinical Pharmacology, University of Bonn, Germany, does not exist any more since more than 15 years and the method described is from a reference dating back to 2003. Please refer to an actual reference and/or method description. What are the main methodological validation scores such as reproducibility, sensitivity, inter- and intra-batch coefficients of variations etc. of this GC-MS method.
2) 2.3 Lipid markers
a) Determination of LDL-C is not mentioned or described.
b) Triglycerides are measured by a method, measuring the amount of total glycerol in the serum. However, this is not exact, because the serum, plasma or blood also contains free glycerol additionally to fatty acid-esterified (triglycerides) glycerides. This free glycerol has to be subtracted from the basic total glycerol amount. Otherwise the true triglyceride levels are overestimated.
c) Cholesterol in serum is measured by the CHOD-PAP method. The authors should state that by this enzymatically photometric method all the sterols with a double bond at position C4 are included in the analysis. Thus all the plant sterols and oxysterols and in part some cholesterol precursors are included in this measurement. So you already correlate plant sterols with itself, included in the cholesterol analysis. Please comment on this. To exclude this bias, cholesterol should be measured from the same sample where the phytosterols were measured by a chromatographic separation method such as GC or LC followed by a detection using flame ionization detection or mass spectrometry or for LC an appropriate uv-method.
3) References
A reference indicating that the data in part or totally have been published in a previous publication in Nutrients 2022, 14, 2500 is lacking.
Additionally pre-publication of the data as an abstract submitted and published in ESC Preventive Cardiology 2022 – ePoster in Epidemiology is not mentioned.
Author Response
Please see attached pdf document

Reviewer 2 Report
Comments and Suggestions for Authors
Dear Editor,
I carefully read the manuscript "Association Between Serum Phytosterols and Lipid Levels in a Population-Based Study".
My comments and suggestions for the authors are the following:
- In the abstract, the authors wrote "Data from 16 910 participants (59.1% women, 70.4±4.7 years) for the first survey and from 721 participants (60.2% women, 75.1±4.7 years) for the second survey were used". Maybe the authors should be more precise in the abstract, as it is not clear which surveys they refer to.
- In the abstract, more quantitative results should be reported.
- Table 1: The abbreviations used in the table should be specified at the bottom of the table.
- "Hypolipidemic drugs" should be replaced by "Lipid-lowering drugs" throughout the manuscript.
- Pag. 3: The authors should specify how the normal distribution of the variables was assessed.
- Study's limitations should be further and more deeply discussed.
- The authors should highly consider to refer to doi: 10.1002/ptr.8052, and doi.org/10.3390/nu15214555 in their manuscript.
Comments on the Quality of English LanguageEnglish language needs to be carefully revised and improved.
Reviewer 3 Report
Comments and Suggestions for Authors
Stanasila et al. assessed the relationship between serum phytosterol levels and lipid values in a community-dwelling Western European population, in two cross-sectional evaluations at a 5-year interval. Most of the research with phytosterols has confirmed their LDL cholesterol lowering effects. However, the results of this study indicate that phytosterols levels are positively associated with total and LDL cholesterol levels, while no consistent associations with other lipid markers were found. The authors should give this seemingly contradictory result more discussion and analysis.
Specific comments
1. Please provide more detailed detection conditions and calculation method of serum phytosterols.
2. Whether a dietary survey was conducted on the subjects to provide dietary phytosterol intake.
3. Please explain the difference between total cholesterol and total cholesterol-GC.
4. It is suggested to analyze the difference of serum phytosterols between different genders.
Comments on the Quality of English LanguageThere are still some minor grammatical/style issues that a copy editor can deal with.
Round 2
Reviewer 2 Report
Comments and Suggestions for Authors
Dear Editor,
I carefully read the revised version of the manuscript, that is significantly improved compared with its original version. I recommend its publication in the Journal.
Author Response
Thank you very much
With regards